# Data Analysis and Machine Learning Applications Curriculum for Open Science

## Abstract

The strong and growing role of machine learning (ML) in the physical sciences is well established and appropriate given the complex detectors and large data sets at the foundational layer of many scientific explorations. Increasingly, Physics departments are offering curricula to their undergraduate and graduate students that focus on the intersection of data science, machine learning and physical science. In this paper, we provide some perspective from experience in developing curriculum at the intersection of ML and physics on the potential role of open science in ML education and present some of the opportunities and challenges in the form of open questions for our community to explore.

## 1. Introduction

It is important to teach students the fundamentals of how to analyze and interpret scientific data. Increasingly, this involves the application machine learning method, tools and techniques to problems common in scientific research such as classification and regression. Each day there are new applications of machine learning to the physical sciences in ways that are advancing our knowledge of nature.

This paper, we provide some perspective from experience in developing curriculum at the intersection of ML and physics on the potential role of open science in ML education and present some of the opportunities and challenges in the form of open questions for our community to explore.

One may ponder the value and effectiveness of physicists teaching ML. Should this not be best left to the computer science departments? It is this question that we explore briefly in this paper.

---

[1]Anonymous Institution, Anonymous City, Anonymous Region, Anonymous Country. Correspondence to: Anonymous Author <anon.email@domain.com>.

Preliminary work. Under review by the Teaching Machine Learning Workshop at ECML 2021. Do not distribute.

## 2. Personal experience

A few years ago, I began discussing with undergraduate physics majors at our university during townhall-style events about our curriculum in an attempt to assess the level of interest in a physics-oriented course in machine learning applications. The response was overwhelmingly positive and it was clear to me that many of our students want this type of training. In 2018, I developed a new course our Department titled *Data Analysis and Machine Learning Applications for Physicists*.

I designed this course to teach the fundamentals of scientific data analysis and interpretation to our students and empower them through practical utilization of modern machine learning tools and techniques using open source data and software. This course has a number of innovative technical elements and was, to my knowledge, the first course in the Physics Department to be delivered solely through Jupyter Notebooks, Git Hub, and mini research projects utilizing *open scientific data*.

I have taught this course to both undergraduate and graduate students over the Spring 2019 and Fall 2019 semesters and continue to develop the curriculum with the help of a postdoc and graduate student in the research group that I lead in particle physics.

This course is designed to be interactive and collaborative, employing Active Learning methods, at the same time developing students' skills and knowledge. We live in an increasingly data-centric world, with both people and machines learning from vast amounts of data. There has never been a time where early-career physicists were more in need of a solid understanding in the basics of scientific data analysis and interpretation, data-driven inference and machine learning, and a working knowledge of the most important tools and techniques from modern data science than today.

Particle physics holds a prominent role within academic curriculum. There are a number of reasons for this, including the "fundamental" nature of our science, the compelling historical develop of our field, theoretical research that applies and develops advanced mathematics, powerful applications, and high-visibility spin-off technologies (e.g. WWW).

At the same time, machine learning has an increasing prominent role in our science, Physicists are increasing collaborating with computer scientists and industry to develop "physics-driven" or "physics-inspired" machine learning architectures and methods.

Topics covered in the course that I mention include:

- Notebooks and numerical python
- Handling and Visualizing Data
- Finding structure in data
- Measuring and reducing dimensionality
- Adapting linear methods to nonlinear problems
- Estimating probability density
- Probability theory
- Statistical methods
- Bayesian statistics
- Markov-chain Monte Carlo in practice
- Stochastic processes and Markov-chain theory
- Variational inference
- Optimization
- Computational graphs and probabilistic programming
- Bayesian model selection
- Learning in a probabilistic context
- Supervised learning in Scikit-Learn
- Cross validation
- Neural networks
- Deep learning

Topics are demonstrated in-class through live-code examples and slides within in Juypter notebooks.

**Lectures** The lectures include physics and data science pedagogy, demonstrated through live examples in Jupyter notebooks that students work through in class.

**Homework** Homework is an important part of the course where students have an opportunity to apply the techniques they learn to problems relevant to the analysis of scientific data. Students submit their homework via a private Github repository they create at the beginning of the semester. The grading is done primarily via NBGRADER but quality is checked and feedback is provided by teaching assistants that grade the submitted material.

**Projects** Approximately halfway through the course, students have the opportunity to choose from a set of projects that use open scientific data. They are asked to answer certain questions about the data, supported by their data analysis and written up in a Jupyter notebook which is submitted

in an analogous manner as the homework. The completed project notebook also should include background information about how the data is generated, its scientific relevance and the students methodology.

Projects in this course that I mention include:

- Searching for Higgs Boson Decays with Deep Learning
- Search for Exotic Particles
- Exploration of the Galaxy Zoo (Sky Survey data)

Addition projects are underway using open data from scientific disciplines such as quantum information, materials science, biophysics, genomics, and others.

## 3. Outlook and Opportunities

There are many opportunities and challenges of developing curriculum at the intersection of physics and machine learning. A crucial element of these courses that involve student projects is the availability and discoverability of *open scientific data for education*. Projects based on ML applications to physics data are a strength of these type of courses. As such, a well-curated set of data for education is critical. Similarly, curating a list of courses that science faculty have developed or taught would be incredibly useful to share information and experience.

Physicists have a great deal to offer in ML education, as topics can be explored using real data from experiments and simulation of phenomena in the world we live.