# OpenReview forum: "Data Analysis and Machine Learning Applications Curriculum for Open Science"
_ecmlpkdd.org/ECMLPKDD/2021/Workshop/TeachML — Submitted to TeachML 2021_

### Official Review · Reviewer_BCU2 · 2021-07-09
**Innovative course, does not focus on AI/ML and paper does not provide enough insights though**

**Rating:** 4
**Confidence:** 5

**Review:**

The paper presents an innovative course that provides physics students with an introduction to data analysis.

Pros:
+ Authors present an innovative course in their department.
+ Coursework problems analyze open scientific data.
+ Discusses how ML adds value to other fields.
+ Presents how physics students can learn data analysis (and ML).

Cons:
- Topics covered are largely statistics, AI/ML is just a minor part.
- Does not include a course evaluation/student feedback.
- Course was taught in 2019. The paper does not discuss how the courses would be offered to students during the COVID-19 pandemic and how the course could permanently include digital learning components.
- Too short, does not provide enough insights.

Authors are advised to resubmit an extended and revised version of this paper to the next edition of this workshop.

---

### Official Review · Reviewer_edzT · 2021-07-16
**a course at the intersection between machine learning and physics**

**Rating:** 5
**Confidence:** 4

**Review:**

The paper describe a course based on personal experience in the intersection field between Physics and Machine Learning.

Pros:
- the idea of teaching ML involving Physicists is interesting
- The use of open scientific data is a plus
- It consider specific and clear machine learning application in physics
- the curated list of machine learning topics

Cons:
- the course has been performed only once in 2019. Has any continuity during covid-19 pandemic?
- the pedagogical and teaching methodology is not clear and is broadly described.
- Lecture subsection is broad and need more details

---

### Decision · Program_Chairs · 2021-07-21

**Decision:**

Reject

**Comment:**

Thank you for submitting this year to the Teaching ML workshop. The reviewers agree that this paper is not ready for publication.

We encourage the authors to keep up their efforts in the field and act upon the suggestions made. We would love to see a submission from you in the future. We cordially invite you dial in for the workshop itself to be part of our community and make contributions there. We are looking forward to hearing from you in the future.